behaviour/ecology/evolution

Batesian, imperfect mimicry, antipredator defence, *Habronattus*, Salticidae, Hymenoptera

**Author for correspondence:**
Lisa A. Taylor
e-mail: lisa.taylor@ufl.edu

# Sexually dimorphic dorsal coloration in a jumping spider: testing a potential case of sex-specific mimicry

Collette Cook[1], Erin C. Powell[1], Kevin J. McGraw[3] and Lisa A. Taylor[1,2]

[1]Department of Entomology and Nematology, and [2]Florida Museum of Natural History, University of Florida, Gainesville, FL 32611, USA
[3]School of Life Sciences, Arizona State University, Tempe, AZ 85287, USA

ECP, 0000-0002-2483-1883; KJM, 0000-0001-5196-6620;
LAT, 0000-0002-0738-4268

To avoid predation, many animals mimic behaviours and/or coloration of dangerous prey. Here we examine potential sex-specific mimicry in the jumping spider *Habronattus pyrrithrix*. Previous work proposed that males' conspicuous dorsal coloration paired with characteristic leg-waving (i.e. false antennation) imperfectly mimics hymenopteran insects (e.g. wasps and bees), affording protection to males during mate-searching and courtship. By contrast, less active females are cryptic and display less leg-waving. Here we test the hypothesis that sexually dimorphic dorsal colour patterns in *H. pyrrithrix* are most effective when paired with sex-specific behaviours. We manipulated spider dorsal coloration with makeup to model the opposite sex and exposed them to a larger salticid predator (*Phidippus californicus*). We predicted that males painted like females should suffer higher predation rates than sham-control males. Likewise, females painted like males should suffer higher predation rates than sham-control females. Contrary to expectations, spiders with male-like coloration were attacked more than those with female-like coloration, regardless of their actual sex. Moreover, males were more likely to be captured, and were captured sooner, than females (regardless of colour pattern). With these unexpected negative results, we discuss alternative functional hypotheses for *H. pyrrithrix* colours, as well as the evolution of defensive coloration generally.

## 1. Introduction

Predation can be dangerous, as many prey species defend themselves with toxins, venom or physical defenses that can harm

(a)    (b)

**Figure 1.** Male *Habronattus pyrrithrix* (*a*) and female *Habronattus pyrrithrix* (*b*). Note the conspicuous dorsal stripe pattern of the male and the cryptic dorsal coloration of the female.

predators (reviewed in [1]). Some undefended prey species take advantage of this and have evolved strategic behaviours, colour patterns and other traits that mimic dangerous prey to deceive and avoid predators [1,2]. Selection to avoid predation has resulted in a striking diversity of mimicry complexes across taxa (e.g. sound mimicry in moths [3], chemical mimicry in mantises [4], flash mimicry in fireflies [5]). Many visual mimics are obvious to our human visual system [6–9], with the mimic bearing a near-perfect resemblance to the model [1]. Recently, there has been growing interest in the phenomenon of imperfect mimicry, where the mimic bears only a slight resemblance to the model [10,11]. Imperfect mimicry occurs more often in nature and is often enhanced by additional mimetic components, such as movement patterns [12–15].

Jumping spiders (family Salticidae) are a highly diverse group that display a wide range of mimicry, most notably the mimicry of ants and mutillid wasps [13,16,17]. These examples often include both morphological mimicry, in which mimics resemble their models in both coloration and general body form, and behavioural mimicry, in which mimics perform characteristic movement patterns typical of their models, such as waving their first pair of legs to appear like antennae [13,18,19]. Jumping spider mimicry ranges from extremely specific and accurate (near-perfect) to seemingly general and imperfect, and males and females often differ in their mimetic strategies [20,21]. Near-perfect Batesian mimicry has been well studied in jumping spiders [13,22–24], yet few studies have examined less accurate (imperfect) mimicry in jumping spiders (but see [19]) or addressed the question of why males use different mimetic strategies than females.

Here we examine a potential case of sex-specific imperfect mimicry in the jumping spider *Habronattus pyrrithrix*. Specifically, we test the hypothesis that different lifestyles and behaviours have led males and females to evolve different dorsal colour patterns in order to avoid predation. *Habronattus pyrrithrix* is a sexually dichromatic jumping spider; males have bright red faces and green front legs that they display to cryptically coloured females during courtship and these colours have been the focus of multiple previous studies [25–28]. However, males and females also differ strikingly in dorsal coloration that is not overtly displayed during courtship (figure 1), and it is these dorsal colour patterns that are the focus of the present study. It has recently been suggested that conspicuous dorsal patterns of males function as deceptive markings by imperfectly mimicking the coloration of aversive hymenopterans, such as the numerous and varied species of wasps and bees that are common in the same habitat [21]. Along with their dorsal coloration, male *Habronattus* also raise and wave their first pair of legs while searching for females; this behaviour has been suggested to enhance the effectiveness of the deceptive signal because the waving legs may subtly resemble moving antennae [21]. This conspicuous colour pattern and behaviour may offer protection for males that are highly active and constantly moving through the leaf litter as they search for females [21]. This is in contrast to cryptic females that spend most of their time foraging using a sit-and-wait strategy [21] and guarding eggs in the leaf litter; in contrast with males, this limited movement of females may make cryptic coloration an ideal strategy.

This hypothesis leads to two experimentally testable predictions: (1) males painted to look like females (with natural male behaviour + artificial female dorsal coloration) should suffer higher predation rates than sham-control males (natural male behaviour + natural male dorsal coloration) and (2) females painted like males (natural female behaviour + artificial male dorsal coloration) should suffer higher predation rates than sham-control females (natural female behaviour + natural female dorsal coloration).

To characterize mimicry and understand the forces that drive its evolution, a receiver that is deceived by the mimic must be identified [29]. Selection driven by invertebrate predators may be an important, yet often overlooked, component of the evolution of mimicry [1]. Previous studies have found that predation pressure from large salticid predators has likely contributed to the evolution of very precise ant mimicry in jumping spiders [22,30]. In some cases, aversions to ants by salticids are clearly innate (e.g. [20,31,32]). Here, we tested the predictions above using colour manipulation of *H. pyrrithrix* with predation experiments in the laboratory using a substantially larger jumping spider, *Phidippus californicus*, as the predator. *Phidippus californicus* is an ideal natural predator for this experiment because, like all jumping spiders, it is highly visual [33] and it co-occurs and is common at our collection sites where it has been seen feeding on *H. pyrrithrix* on numerous occasions (LA Taylor, personal observation, 2005–2020). Whether they are hunting *Habronattus* from above (as they are moving through vegetation) or hunting on a flat surface such as the ground, *Phidippus* would be able to clearly view *Habronattus* dorsal patterns; as such, these patterns are well positioned to act as a deterrent to predation in this context.

# 2. Methods

## 2.1. Predation experiment

We collected mature adult *H. pyrrithrix* ($n = 56$ total; 28 adult females, 28 adult males) and juvenile *P. californicus* ($n = 28$) from an agricultural field in Queen Creek, AZ (Maricopa County), USA. All *P. californicus* were collected as spiderlings (between 2 and 3 mm in length) and reared in the laboratory (the number of predators collected here is higher than the total number of trials reported below; we collected excess predators to ensure that enough reached the appropriate size and were willing to participate in our tests). Because they were collected at this small size, we knew that they had not yet eaten adult *H. pyrrithrix* and were therefore naive to predatory interactions with them. The *P. californicus* were fed hatchling crickets (*Acheta domesticus*) three times per week until they had reached a minimum length of 10 mm. Once they had reached this size but were not yet mature, they were ready to participate in experiments. When used in experiments, the *P. californicus* were about twice the body length of the *H. pyrrithrix* (mean total body length in mm ($\pm$s.e.): *P. californicus*=11.87 $\pm$ 0.22, *H. pyrrithrix* males = 5.27 $\pm$ 0.15, *H. pyrrithrix* females = 6.64 $\pm$ 0.14).

We housed *Habronattus pyrrithrix* in the laboratory until experiments were conducted. Before experiments began, all spiders were photographed from above next to a size standard to measure their carapace width; because these spiders do not molt after maturity, this measure of body size is fixed at maturity and therefore does not change throughout the experiment. Spiders were grouped by collection date, ranked by body size (males and females ranked separately by carapace width) and then assigned to groups of four (tetrads of two males and two females). They were ranked by body size to reduce variation in size between spiders in each tetrad. Within each tetrad, one male and one female were randomly assigned to the treatment group (to have their dorsal coloration manipulated; see below) and the other male and female were assigned to the control group (to be sham-manipulated). Each tetrad was then randomly assigned to an individual predator (*P. californicus*). Because the goal of the experiment was to determine how colour and behaviour interact to influence predation under natural conditions (where *H. pyrrithrix* populations are often dense and conspecifics interact frequently with one another [34]), each trial consisted of a single *P. californicus* predator and one tetrad of potential prey spiders (*H. pyrrithrix*; $n = 14$ trials).

## 2.2. Colour manipulation

We manipulated dorsal colour of *H. pyrrithrix* (as described below) 24 h before individuals appeared in an experimental trial (in random order within each tetrad). Individuals were immobilized with $CO_2$ for approximately 3 min while their colour was manipulated and were allowed to recover overnight. To manipulate the back pattern of females to appear male-like, we painted the dorsal carapace and

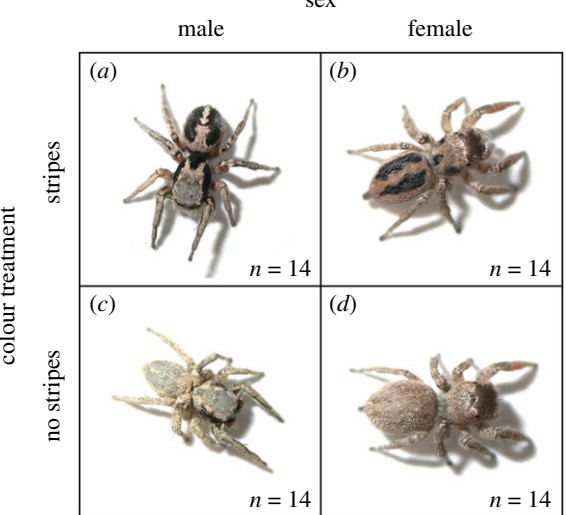

**Figure 2.** Experimental colour manipulation showing (*a*) a sham-control male (male behaviour + male coloration), (*b*) a female painted like a male (female behaviour + male coloration), (*c*) a male painted like a female (male behaviour + female coloration) and (*d*) a sham-control female (female behaviour + female coloration).

abdomen with male-like dorsal stripe patterns using black liquid eyeliner (colour: 'Perversion,' Urban Decay Cosmetics, Costa Mesa, CA; figure 2*a,b*). To similarly manipulate male spiders to appear female-like, we concealed the male dorsal stripe pattern by applying tan foundation powder to the dorsal carapace and abdomen (bareMinerals foundation, colour: 'Light,' Bare Escentuals, San Francisco, CA, USA; figure 2*c,d*). We used these specific brands and types of makeup because they have been successfully used in multiple colour manipulation experiments with *H. pyrrithrix* and they showed no adverse effects on spider behaviour [26,28]. Moreover, the black eyeliner and the foundation powder have reflectance properties that are similar to black markings of males and dorsal coloration of females, respectively (figure 2; spectral properties of this makeup are provided in [26,28]). Both have low reflectance in the UV and lack UV peaks; as such, they should not create unwanted artefacts for predators with UV vision. To control for any unintended effect of the makeup (e.g. olfactory cues), control spiders were also sham-treated with eyeliner and/or foundation applied to the underside of their abdomen (where it is not visible to predators; figure 2*a,d*).

It is important to note that we only manipulated dorsal coloration here; this meant that other morphological differences between males and females remained intact. This included not only behaviour, but also the males' courtship coloration (red faces and green front legs that are typically hidden from predators) as well as sex differences in body size and allometry (e.g. the relatively larger abdomens of females compared with males, figure 1). This design allowed us to examine how dorsal patterns function when combined with sex-specific behaviour (alongside other aspects of sex-specific morphology).

## 2.3. Experimental trials

The experimental predation chamber consisted of a clear plastic box (21 × 17 × 22.5 cm) with a clear Plexiglas lid. This chamber was surrounded by a larger plastic container filled with 1 cm of sand at the bottom and leaves (50/50 mixture of cottonwood leaf litter (*Populus fremontii*) and desert willow leaf litter (*Chilopsis linearis*)). This set-up allowed the spiders to move around freely within the chamber while being videotaped from above. The surrounding sand and leaf litter provided a natural visual background similar to the background the spiders would encounter in the field; because the leaf litter was outside the predation chamber, it did not obstruct the camera's view of interacting spiders (figure 3). A *Phidippus californicus* predator was placed in a clear plastic vial (10 cm tall and 3 cm in diameter) in the centre of the experimental chamber the night before the test in order to acclimate. At the time of the test, the four *H. pyrrithrix* were placed in separate vials (6.5 cm tall and 2.5 cm in diameter) in each corner of the predation chamber, while the predator remained in the larger vial in the centre of the chamber (figure 3). All spiders were allowed to acclimate for 10 min in the chamber, after which the lids of the vials of *H. pyrrithrix* were quickly removed in random order and the lid of the predator was subsequently removed, allowing all spiders to exit the vials and roam

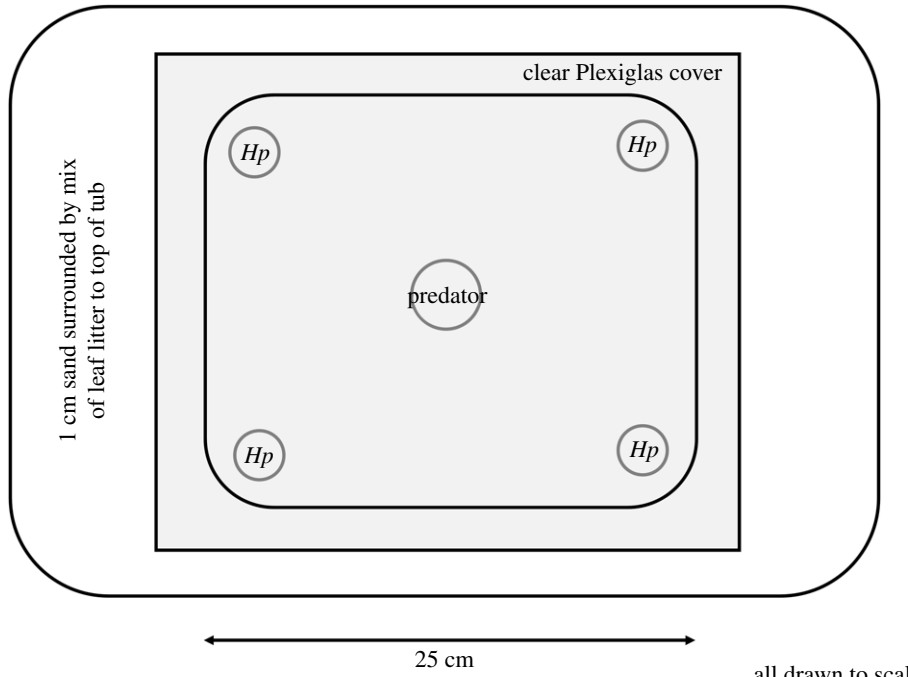

**Figure 3.** The experimental chamber for running predator–prey trials. Four *H. pyrrithrix* were released from vials in each corner of the predation chamber, while the *P. californicus* was released from a larger vial in the centre of the chamber. A natural visual background of sand was placed below the clear floor of the chamber, with leaf litter surrounding all sides.

freely throughout the experimental chamber. All five spiders (the predator and the four *H. pyrrithrix*) were allowed to interact freely throughout the trial.

We recorded all behaviours using video cameras mounted above the chamber for 2 h; this timeframe allowed us to see which *H. pyrrithrix* spiders were captured by the predator, and the order of capture. A 'capture' was defined as a predator successfully attacking, immobilizing and eating an *H. pyrrithrix*. During the first 30 min of each trial (or until the first spider was captured by the predator, whichever came first), an observer watched the trial in real time and recorded the following behaviours: (1) amount of time the predator spent staring at each spider (i.e. predator was stationary and had its forward-facing anterior median eyes directly oriented towards a prey spider), (2) amount of time the predator spent stalking each spider (i.e. predator oriented its body towards prey, crouched down and slowly approached it; this behaviour almost always precedes an attack (LA Taylor, personal observation, 2005–2020), and (3) number of attacks (i.e. predator lunges or jumps directly at a prey spider). Direct observations were used in addition to the videos to ensure that no behaviours were missed (e.g. if a spider was obscured from the video because they were behind part of the vial or another spider). If there were no attacks in the first 30 min of a trial (4 out of 14 trials), the predator was removed, all *H. pyrrithrix* were returned to their vials in the corners of the arena, a new *P. californicus* predator was introduced into the centre vial, and the trial began again. Upon successful completion of a trial (that included at least one attack from the predator on one prey spider), no spiders from that trial were used in any subsequent trials.

## 2.4. Statistical analysis

All analyses were conducted in JMP Pro 15.0, except for the GLMM (see below) which was done in SPSS 2019.

Based on our focal hypothesis, we predicted that we would find an interaction between sex and dorsal pattern in all of the models described below (with male spiders eluding predators better when they have stripes and female spiders eluding predators better when they have no stripes).

We used analyses of variance (ANOVA) to examine how *H. pyrrithrix* sex, colour treatment (dorsal stripes or no stripes), and their interaction affected the time the predator spent staring at or stalking each spider, and the number of attacks each spider received from the predator. The number of attacks (count data) were rank-transformed (following the RT-1 method in Conover & Iman [35]) because

**Table 1.** Results of ANOVAs showing effects of *H. pyrrithrix* sex, colour treatment and their interaction on time that the predator spent staring at and stalking each spider and the number of attacks each spider received from the predator. Regardless of their actual sex, spiders with male-like dorsal patterns (stripes) were attacked more than those with female-like dorsal coloration (no stripes). Significant *p*-values are shown in italics.

|  | F | d.f. | p |
|---|---|---|---|
| time spent staring by predator | | | |
| sex | 0.036 | 1,36 | 0.85 |
| colour treatment | 0.39 | 1,36 | 0.53 |
| sex × colour treatment | 0.001 | 1,36 | 0.98 |
| time spent stalking by predator | | | |
| sex | 0.0061 | 1,36 | 0.94 |
| colour treatment | 3.13 | 1,36 | 0.086 |
| sex × colour treatment | 0.17 | 1,36 | 0.68 |
| number of attacks from predator | | | |
| sex | 0.39 | 1,36 | 0.53 |
| colour treatment | 4.48 | 1,36 | *0.041* |
| sex × colour treatment | 0.59 | 1,36 | 0.45 |

they did not meet assumptions of parametric statistics. Because trials consisted of tetrads of *H. pyrrithrix* with a single predator, trial ID was included as a random factor in all models. Data for these models came from observation periods that varied in length (depending on how soon the predator in each trial made its first attack; see Methods); we accounted for this using trial ID as a random factor. One trial was excluded from these analyses because the predator leapt out of its starting vial and immediately attacked its first spider; this meant that the observation period ended immediately upon starting, resulting in no data for staring, stalking or attacks on any of the other spiders in the arena.

We also used a generalized linear mixed model (GLMM) with a binomial distribution to examine how sex, colour treatment, and their interaction affected whether or not each spider was captured by the predator during the 2 h trial. Again, trial ID was included as a random factor. We used ANOVA to examine how sex, colour treatment and their interaction affected the order of each spider's capture within the 2 h trials (with capture order being ranks within each trial, following the RT-2 method in Conover & Iman [35]). Because capture-order data were ranks within trials, trial ID was not included as a random factor in this model.

# 3. Results

Contrary to our predictions, we found no significant interactions between sex and colour treatment in any of our analyses. Neither sex, colour treatment, nor their interaction affected the amount of time the predator spent staring at the spiders or the amount of time spent stalking them, although there was a non-significant trend towards spiders with male-like dorsal patterns (stripes) to be stalked more by predators than those without (table 1). Regardless of their actual sex, spiders with male-like dorsal patterns (stripes) were attacked significantly more than those with female-like dorsal patterns (no stripes; table 1 and figure 4). Regardless of whether or not the spider had dorsal stripes (male-like colour patterns) or no dorsal stripes (female-like colour patterns), males were more likely to be captured than females (table 2 and figure 5). Moreover, males (regardless of colour pattern) were captured sooner than females (table 3).

# 4. Discussion

Our *a priori* hypothesis was that the sexually dimorphic dorsal colour patterns in *Habronattus pyrrithrix* are most effective at deterring predators when paired with sex-specific behaviours. As such, in our experiments we expected to see an interaction between the sex of the spider and their dorsal colour

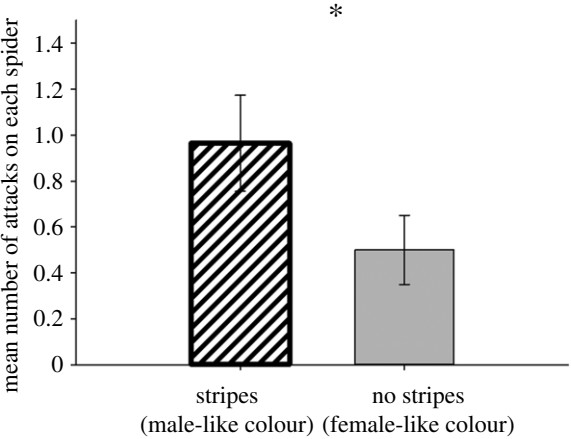

**Figure 4.** Regardless of their actual sex, spiders with male-like dorsal patterns (stripes) were attacked more by the predator than those with female-like patterns (no stripes). An asterisk indicates a significant difference between treatment groups.

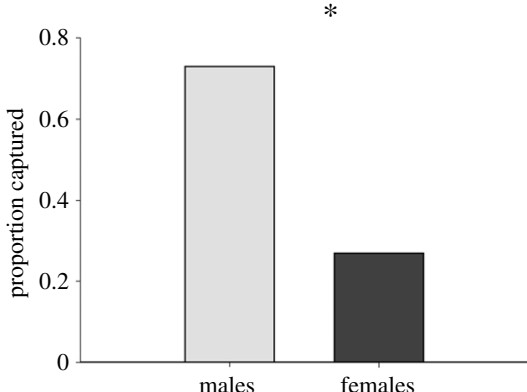

**Figure 5.** Regardless of their dorsal patterns (stripes or no stripes), males were more likely to be captured by the predator than females. An asterisk indicates a significant difference between groups.

**Table 2.** Results of GLMM showing how sex, colour treatment and their interaction affects the likelihood of being captured. Regardless of their dorsal pattern, males were more likely to be captured than females. Significant *p*-values are shown in italics.

|  | *F* | d.f. | *p* |
| --- | --- | --- | --- |
| sex | 9.59 | 1,52 | *0.003* |
| colour treatment | 0.002 | 1,52 | 0.96 |
| sex × colour treatment | 0.35 | 1,52 | 0.56 |

**Table 3.** Results of ANOVA showing how sex, colour treatment and their interaction affected the order in which spiders were captured during trials. Regardless of their dorsal pattern, males were captured sooner than females. Significant *p*-values are shown in italics.

|  | *F* | d.f. | *p* |
| --- | --- | --- | --- |
| sex | 9.27 | 1,52 | *0.0037* |
| colour treatment | 0.077 | 1,52 | 0.78 |
| sex × colour treatment | 0.31 | 1,52 | 0.58 |

pattern (stripes or no stripes), with dorsal stripes conferring an advantage to males and cryptic coloration (no stripes) conferring an advantage to females. However, our results did not match these predictions. Instead, regardless of the actual sex of the spider, having a male-like dorsal pattern (stripes) increased the rate that an *H. pyrrithrix* individual would be attacked by *P. californicus* predators. We also found that, regardless of their dorsal pattern, male *H. pyrrithrix* were more likely than females to be captured by the predator and were captured sooner than females. These results offer no support for our hypothesis, at least in the specific context of interactions with this particular predator. This leaves us with two possible interpretations of our data. The first possibility is that our hypothesis for the function of these colour patterns is still valid, but that these functions only work to deter predation by other predators, not the relatively naive *P. californicus* examined here. Indeed, previous work with *H. pyrrithrix* has shown that predation comes from many sources [21], including aerial visual predators that might also be influenced by dorsal colour patterns. A second possibility is that these sexually dimorphic dorsal colour patterns have evolved for a different reason. It is easy to explain why females are cryptic, as they have limited movement compared to males [21] and cryptic coloration works best when animals are stationary [36]. The conspicuous striped markings of males (that are common across the genus *Habronattus* [21,37]) warrant more of an explanation, particularly because our results here suggest that they increase, rather than decrease, attacks from at least one important *Phidippus* predator. Here we discuss possibilities such as disruptive or motion dazzle functions for these colours as well as the possibility that they serve no function at all.

In light of our data, we should first explore the possibility that, although *P. californicus* (and other large species of jumping spiders) are common and abundant and are likely important predators of *H. pyrrithrix*, they may not be the predators driving their sexually dimorphic colour patterns. We expected *P. californicus* to be deterred by male-like dorsal stripe patterns paired with male-like behaviour (i.e. leg-waving) because, to our eyes, this combination subtly resembles hymenopterans [21]. However, like all jumping spiders, *Phidippus* has impressive visual acuity [33]. Indeed, some large jumping spiders are able to visually distinguish between ants and extremely accurate ant mimics [30]; if *Phidippus* are also capable of this, it may be that they are not deceived by the vague, imperfect mimicry hypothesized here. Alternatively, it may be that a hymenopteran-like appearance (even an effective one) is simply not as strong a deterrent to *Phidippus* as we expected. When given the choice, we would expect *Phidippus* to avoid potentially dangerous prey such as hymenopterans, but we know that they will sometimes feed on honeybees and other small hymenopterans in both the laboratory and field (LA Taylor, personal observation, 2005–2020). Another possibility is that we would have seen the expected responses from *Phidippus* predators if they had more opportunities to develop learned hymenopteran aversions in the field. It was important for us to use *Phidippus* raised in the laboratory for our experiment so that we could be sure that they had no prior experience eating *Habronattus*. However, this meant that they also had limited opportunities to interact naturally with hymenopterans before being collected for our study. While we know that aversions to hymenopterans can be innate in salticids [20,31,32], we also know that *Phidippus* can learn prey aversions from experience (e.g. [38,39]). The receiver psychology, sensory ecology and perceptual biases of predator species undoubtedly affects the evolution of mimicry, and these factors should be considered in the interpretation of our results and in follow-up work on mimicry in salticids [29].

If data from *Phidippus* does not support our hypothesis, might it still hold up with data from other predators? There are clearly limits to what we can conclude from a study of just one predator and there are several other visual predators that are common, abundant, and feed readily on *H. pyrrithrix* that should be considered [21]. For males, this includes cannibalistic conspecific females and even heterospecific *Habronattus* females that are the targets of misdirected courtship in the field [34]. As has been suggested elsewhere [21], the striped dorsal patterning on males, combined with their characteristic leg-waving behaviour, may afford them some protection from the very females that they are courting (both conspecifics and heterospecifics); even if these dorsal colours are not explicitly displayed to females, they still may protect them from a female who approaches them from behind. Like *Phidippus*, these *Habronattus* females also have good visual acuity, but due to their smaller size they may be less likely to attack hymenopteran-like prey that is perceived to be risky.

There are several additional predators that should be considered for tests of our hypothesis. For example, it would be informative to repeat our experiment with visual predators documented to be averse to hymenopteran prey, such as mantises [40]. As our focus is on dorsal colour patterns, we might also want to consider aerial predators that search the ground for prey, such as birds, dragonflies, damselflies and mud dauber wasps that are common in their riparian habitat. Indeed, naive birds are less likely to eat prey paired with high contrast stripes [41]. Further, we know that

dragonflies avoid hymenopterans [42] and, in some experiments, they will actively avoid prey to which stripes have been added [43]. Perhaps most interesting are the *Sceliphron caementarium* mud dauber wasps that are common and abundant at our field sites. They specialize broadly but exclusively on spiders, they never feed on hymenopterans, and individuals specialize on particular spider types [44]. The dorsal stripe patterns of male *H. pyrrithrix* combined with male leg-waving behaviour might simply allow these males to escape the spider-specific search image of a hunting *Sceliphron*. Imperfect mimicry, such as what we propose here, can result from having a diverse suite of predators spanning various taxa if each predator responds differently to prey [19,45,46]. Future work should clearly consider a larger suite of predators to achieve a more holistic view of the selective pressures these spiders face in their natural environment.

In addition to extending tests of our current focal hypothesis, we should also consider alternative functional hypotheses for the striped coloration of males, as well as non-adaptive explanations. Disruptive coloration is a form of camouflage that uses contrasting markings to break up an animal's outline; these typically occur along borders and edges and function to hinder recognition of an animal's true shape [47–49]. As we have argued elsewhere [21], the dorsal patterns of male *H. pyrrithrix* examined in our study do not appear to fit this description, as the stripes are in the interior of the dorsal abdomen and carapace and they are encircled with black and white in a way that enhances, rather than disrupts, the outline of the body (figure 1a). Moreover, the constant movement and leg-waving of males attracts attention to the head of the spider, making it quite easy, at least for human observers, to spot (and collect) these spiders in large numbers in the field (LA Taylor, personal observation, 2005–2020). Motion dazzle is another possible function for striped coloration, as such colour patterns might make it difficult for predators to assess a fleeing animal's speed and trajectory at high speeds [50–52]. Although this might explain striped patterns in some fast-moving animals such as snakes (see [53]), it seems less likely in *H. pyrrithrix* where fleeing spiders move more slowly and generally don't move in a straight line, but alternate between leaping and stopping as they zigzag through the leaf litter (LA Taylor, personal observation, 2005–2020). While we argue neither disruptive coloration nor motion dazzle functions are likely, we cannot discount them completely and should continue to pursue these ideas with future experiments using different predators. We should also remember to consider the possibility that male colour patterns may not have an adaptive function at all [54].

Despite the lack of evidence for our focal hypothesis and the need for more studies to address it further, our findings from the present study provide some interesting insights into the biology of *H. pyrrithrix*. One particularly interesting finding was that males were captured more by the predator than females, even though the sexes were attacked at equal rates. This suggests that females may be better able to avoid capture when they face attack by a predator. Male *H. pyrrithrix* spend significant time courting females in the field [34] and in the laboratory [26,28] and when they do so, they appear to be intensely focused on courtship, often ignoring their surroundings (LA Taylor, personal observation, 2005–2020). In the present study, all four spiders in a trial were allowed to freely interact as they do frequently in nature [34], which meant that males could actively court females. The intense focus by males on courtship may have resulted in males being less vigilant, less perceptive of predators, and therefore less likely to escape when attacked. In nature, even when males are not actively courting females, they spend significant amounts of time moving through the leaf litter [21]; this may also contribute to lower vigilance compared with more stationary females. Males and females also differ in size and allometry; it may be that the smaller males are less able to defend themselves (compared with larger and heavier females) and are therefore more easily captured.

A second interesting and unexpected finding was that dorsal stripes (having a male-like colour pattern) increased attack rate regardless of the actual sex of the spider. Though we expected black and white stripes on male spiders to be avoided by *P. californicus*, we may have seen the increased attacks on striped spiders, in part, because the highly contrasting stripes are so conspicuous and increased visual detection. Indeed, previous work by our laboratory group with the congener, *Phidippus regius*, suggests that black and white stripes are highly conspicuous: in predation experiments, *P. regius* oriented more often and faster to small colour-manipulated termite prey with bold black and white stripes compared to uniformly white termites (L Gawel, M Brock, L Taylor, unpublished data, 2015–2016). This pattern of increased attention to striped prey also held in similar predation experiments with another jumping spider (*Maevia inclemens* [55]). However, neither of these previous studies found higher attack rates on striped prey as we did here (L Gawel, M Brock, L Taylor, unpublished data, 2015–2016 [55]). More work is needed to understand how bold patterns such as stripes, and the context in which they are encountered, influence predator attention, prey selection or prey aversion. In our experiment, it is not clear why the increased attack rate on striped individuals did not translate directly to an increased capture rate. One possibility

that should be examined further is that predators attacking striped prey do so more cautiously, in ways that allow the prey spider to increase their chances of escape [56].

In conclusion, the reason why male and female *H. pyrrithrix* differ so drastically in dorsal coloration remains elusive. Here we present negative results for our focal hypothesis and ideas for moving this work forward. It has long been known that the field of biology (including ecology and evolution) suffers from a 'file drawer problem', where a majority of non-significant results go unpublished [57]. More recent work suggests that this problem is increasing across many fields of science, limiting scientific progress [58,59]. We hope that the unexpected negative results presented here will be useful not just for understanding *H. pyrrithrix*, but for generating ideas and improving our understanding of animal colour patterns more generally.

Data accessibility. Data available from the Dryad Digital Repository: https://doi.org/10.5061/dryad.zpc866t7t [60].
Authors' contributions. L.A.T. and K.M. designed the study; L.A.T. and C.C. collected the data; C.C., E.C.P. and L.A.T. analysed the data; C.C., E.C.P. and L.A.T. prepared the figures; C.C., E.C.P., K.M. and L.A.T. wrote the manuscript. All authors read and approved the final manuscript.
Competing interests. The authors declare no competing interests.
Funding. C.C. was supported by the Research Experiences for Undergraduates program from the National Science Foundation (NSF grant no. IOS-1557867 to L.A.T.). E.C.P. was supported by a Career Life Balance supplement from NSF (and grant nos. IOS-1557867 and IOS-1831751 to L.A.T.). Funding was also provided by an NSF Graduate Research Fellowship (to L.A.T. during the early stages of this study).
Acknowledgements. We thank all members of the Taylor and McGraw laboratories for feedback on study design and discussion. We thank G.B. Edwards and S. Brown for identifying the *P. californicus*.

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
