## [Peer Review File · Royal Society Open Science]

Review History

RSOS-210308.R0 (Original submission)

Review form: Reviewer 1

Is the manuscript scientifically sound in its present form?

Yes

Are the interpretations and conclusions justified by the results?

Yes

Is the language acceptable?

Yes

Do you have any ethical concerns with this paper?

No

Have you any concerns about statistical analyses in this paper?

No

Recommendation?

Accept with minor revision (please list in comments)

Comments to the Author(s)

General Comments:

This is an interesting study, though the authors obtained negative results, which don't support their a priori hypothesis. I agree with the authors that journals should have tried to accept those studies with negative results as since "the unexpected negative results ...but for generating ideas and improving our understanding of animal color patterns more generally'. This study is the case which generates 'ideas for moving the work forward', I think. Although the study provides no evidence that male color pattern+sex-specific behavior confers an advantage to males that are proposed to imperfectly mimicking wasps, the hypotheses/predictations are very clearly spelt out, the experiments are accordingly well designed and nicely performed, subsequent data analyses are also well done and presented. Thus, I think it is scientifically sound. Authors then provide explanations for their results, offer alternative functional hypotheses for striped color pattern of males such as disruptive and motion dazzle functions as well as even no function at all. Authors even provide two interesting insights into the biology of the species tested. I think this paper sets a stand for how to present a study with negative results for many journals, particularly those insights for moving the research forward. The manuscript is well written and presented. My only concern is that the discussion may be too long, which could be shortened one third, though many explanations could be provided for the negative results.

Specific Comments:

p2, line 26, Abstract: define 'dangerous prey' and also add 'toxic or before 'dangerous prey'.

p2, line 29, Abstract: specify what are the possible hymenopteran insects.

p2, line 34, Abstract: spell out the species name of the predator.

p3, line 49: again define 'dangerous prey'.

p3, line 50: add a full stop '.' Before 'Selection to avoid ...'.

P6, lines 117-118: provide the data (mean/se) on body size of both *H. pyrrithrix* (males and females) and *P. californicus* that were used in the tests.

P9, lines 184-195: is there any record of activity level for both males (2) and females (2) in each trial, which may be important for attention of predators.

P12-15, lines 260-311: this part about possible predators for testing is too long, which can be shortened a lot.

P15, lines 332-344: That females are less likely captured than males could not be due to how they are able to better avoid capture, but also may be because they are not that active. In addition, if the size of males and females is a possible factor, you could include body size of all 4 spiders as a predictor in the models.

Review form: Reviewer 2

Is the manuscript scientifically sound in its present form?

Yes

Are the interpretations and conclusions justified by the results?

Yes

Is the language acceptable?

Yes

Do you have any ethical concerns with this paper?

No

Have you any concerns about statistical analyses in this paper?

No

Recommendation?

Accept with minor revision (please list in comments)

Comments to the Author(s)

Dear Editor and authors:

This is a simple, yet elegant study that addresses the question of whether dorsal patterns in a jumping spider have evolved in males as Batesian mimicry, due to males being more exposed to predation. The manuscript is well written, the objectives clearly presented, and the methods are explicit and easy to replicate. Although I personally prefer not to transform count data and instead use negative binomial regression models, the data was correctly transformed for the analyses, and so I have no comments regarding analyses. Against the predictions, the male coloration did not confer any advantage when exposed to a predator, and instead, individuals with the male markings (males and painted females) received more attacks compared to individuals without male markings (females and painted males). Regardless of coloration, males were more likely to be predated. A positive result would have provided evidence for the Batesian mimicry hypothesis, however a negative result does not provide evidence against it, as many other factors (i.e. other predators) were not included in this study. I am pleased with the way the authors acknowledge this in the discussion.

The weakness of this study is that the predation hypothesis was tested with a single predator. However, this has been thoroughly discussed, including the suggestion of specific predators for subsequent similar studies to test the Batesian mimicry hypothesis. Since jumping spiders can see in the UV spectrum, I wonder how that may have influenced the results of this study, and I would suggest the authors to expand on this in the discussion – a sentence or two would be enough. In addition, the number of trials is missing in the manuscript. Since the authors collected 28 *P. californicus* spiders (predators), but one experiment was eliminated, one can easily assume that the total of the trials is 27, but in the data files it's clear that there were 13 analysed trials. Other than that, I have no further comments. The authors did an excellent job in carefully editing the manuscript and I found no mistakes.

Decision letter (RSOS-210308.R0)

Dear Dr Taylor

On behalf of the Editors, we are pleased to inform you that your Manuscript RSOS-210308 "Sexually dimorphic dorsal coloration in a jumping spider: testing a potential case of sex-specific mimicry" has been accepted for publication in Royal Society Open Science subject to minor revision in accordance with the referees' reports. Please find the referees' comments along with any feedback from the Editors below my signature.

Please submit your revised manuscript and required files (see below) no later than 7 days from today's (ie 19-May-2021) date. Note: the ScholarOne system will 'lock' if submission of the revision is attempted 7 or more days after the deadline. If you do not think you will be able to meet this deadline please contact the editorial office immediately.

on behalf of Dr Kimberley Mathot (Associate Editor) and Kevin Padian (Subject Editor)
openscience@royalsociety.org

Associate Editor Comments to Author (Dr Kimberley Mathot):
Comments to the Author:
Dear Dr. Taylor,

I apologize for the delay in reaching a decision, but it took longer than expected to find suitable referees. We have now received referee reports from two experts in your field. They are both very positive about this study, and I agree with their assessment. The hypothesis and predictions were clearly laid out, and the study design to test the hypothesis was appropriate and described in good detail. The post-hoc interpretation of the unexpected results was also appreciated by myself and the referees.

The referees raise only a small number of very minor points to be addressed, and I am therefore recommending your manuscript be accepted with minor revisions. Although referee #1 recommends shortening the discussion, I don't find it overly long. If you are able to shorten it by being more concise, that would be appropriate, but I would not like to see it shortened at the cost of losing content. In fact, referee #2 suggests an addition to the discussion (a brief discussion of how the fact that jumping spider see in the UV spectrum might have influenced the results). I agree this would be a good addition, and likely only requires an additional sentence or two.

Thank you for submitting your work for consideration at Royal Society Open Science.

Reviewer comments to Author:

Reviewer: 1

Comments to the Author(s)

General Comments:

This is an interesting study, though the authors obtained negative results, which don't support their a priori hypothesis. I agree with the authors that journals should have tried to accept those studies with negative results as since "the unexpected negative results ...but for generating ideas and improving our understanding of animal color patterns more generally'. This study is the case which generates 'ideas for moving the work forward', I think. Although the study provides no evidence that male color pattern+sex-specific behavior confers an advantage to males that are proposed to imperfectly mimicking wasps, the hypotheses/predictations are very clearly spelt out, the experiments are accordingly well designed and nicely performed, subsequent data analyses are also well done and presented. Thus, I think it is scientifically sound. Authors then provide explanations for their results, offer alternative functional hypotheses for striped color pattern of males such as disruptive and motion dazzle functions as well as even no function at all. Authors even provide two interesting insights into the biology of the species tested. I think this paper sets a stand for how to present a study with negative results for many journals, particularly those insights for moving the research forward. The manuscript is well written and presented. My only concern is that the discussion may be too long, which could be shortened one third, though many explanations could be provided for the negative results.

Specific Comments:

p2, line 26, Abstract: define 'dangerous prey' and also add 'toxic or before 'dangerous prey'.

p2, line 29, Abstract: specify what are the possible hymenopteran insects.

p2, line 34, Abstract: spell out the species name of the predator.

p3, line 49: again define 'dangerous prey'.

p3, line 50: add a full stop '.' Before 'Selection to avoid ...'.

P6, lines 117-118: provide the data (mean/se) on body size of both *H. pyrrithrix* (males and females) and *P. californicus* that were used in the tests.

P9, lines 184-195: is there any record of activity level for both males (2) and females (2) in each trial, which may be important for attention of predators.

P12-15, lines 260-311: this part about possible predators for testing is too long, which can be shortened a lot.

P15, lines 332-344: That females are less likely captured than males could not be due to how they are able to better avoid capture, but also may be because they are not that active. In addition, if the size of males and females is a possible factor, you could include body size of all 4 spiders as a predictor in the models.

Reviewer: 2
Comments to the Author(s)
Dear Editor and authors:

This is a simple, yet elegant study that addresses the question of whether dorsal patterns in a jumping spider have evolved in males as Batesian mimicry, due to males being more exposed to predation. The manuscript is well written, the objectives clearly presented, and the methods are explicit and easy to replicate. Although I personally prefer not to transform count data and instead use negative binomial regression models, the data was correctly transformed for the analyses, and so I have no comments regarding analyses. Against the predictions, the male coloration did not confer any advantage when exposed to a predator, and instead, individuals with the male markings (males and painted females) received more attacks compared to individuals without male markings (females and painted males). Regardless of coloration, males were more likely to be predated. A positive result would have provided evidence for the Batesian mimicry hypothesis, however a negative result does not provide evidence against it, as many other factors (i.e. other predators) were not included in this study. I am pleased with the way the authors acknowledge this in the discussion.

The weakness of this study is that the predation hypothesis was tested with a single predator. However, this has been thoroughly discussed, including the suggestion of specific predators for subsequent similar studies to test the Batesian mimicry hypothesis. Since jumping spiders can see in the UV spectrum, I wonder how that may have influenced the results of this study, and I would suggest the authors to expand on this in the discussion – a sentence or two would be enough. In addition, the number of trials is missing in the manuscript. Since the authors collected 28 *P. californicus* spiders (predators), but one experiment was eliminated, one can easily assume that the total of the trials is 27, but in the data files it's clear that there were 13 analysed trials. Other than that, I have no further comments. The authors did an excellent job in carefully editing the manuscript and I found no mistakes.

===PREPARING YOUR MANUSCRIPT===

If you have been asked to revise the written English in your submission as a condition of publication, you must do so, and you are expected to provide evidence that you have received language editing support. The journal would prefer that you use a professional language editing

service and provide a certificate of editing, but a signed letter from a colleague who is a native speaker of English is acceptable. Note the journal has arranged a number of discounts for authors using professional language editing services (<https://royalsociety.org/journals/authors/benefits/language-editing/>).

===PREPARING YOUR REVISION IN SCHOLARONE===

-- If you have uploaded ESM files, please ensure you follow the guidance at <https://royalsociety.org/journals/authors/author-guidelines/#supplementary-material> to

include a suitable title and informative caption. An example of appropriate titling and captioning may be found at https://figshare.com/articles/Table_S2_from_Is_there_a_trade-off_between_peak_performance_and_performance_breadth_across_temperatures_for_aerobic_scope_in_teleost_fishes_/3843624.

Author's Response to Decision Letter for (RSOS-210308.R0)

See Appendix A.

Decision letter (RSOS-210308.R1)

Dear Dr Taylor,

I am pleased to inform you that your manuscript entitled "Sexually dimorphic dorsal coloration in a jumping spider: testing a potential case of sex-specific mimicry" is now accepted for publication in Royal Society Open Science.

on behalf of Dr Kimberley Mathot (Associate Editor) and Kevin Padian (Subject Editor)
openscience@royalsociety.org

Appendix A

Dear Dr. Mathot,

We appreciate the positive and constructive comments and suggestions and believe that they have improved the manuscript. Specifically, we addressed UV vision and added size ranges for the focal spiders, among other minor changes. Below you will find our point-by-point responses to each comment. Please let us know if you have any additional concerns or suggestions.

Sincerely,

Lisa Taylor

Comments from Associate Editor (Dr Kimberley Mathot):

Dear Dr. Taylor,

I apologize for the delay in reaching a decision, but it took longer than expected to find suitable referees. We have now received referee reports from two experts in your field. They are both very positive about this study, and I agree with their assessment. The hypothesis and predictions were clearly laid out, and the study design to test the hypothesis was appropriate and described in good detail. The post-hoc interpretation of the unexpected results was also appreciated by myself and the referees.

The referees raise only a small number of very minor points to be addressed, and I am therefore recommending your manuscript be accepted with minor revisions. Although referee #1 recommends shortening the discussion, I don't find it overly long. If you are able to shorten it by being more concise, that would be appropriate, but I would not like to see it shortened at the cost of losing content. In fact, referee #2 suggests an addition to the discussion (a brief discussion of how the fact that jumping spider see in the UV spectrum might have influenced the results). I agree this would be a good addition, and likely only requires an additional sentence or two.

Thank you for submitting your work for consideration at Royal Society Open Science.

Reviewer: 1

This is an interesting study, though the authors obtained negative results, which don't support their a priori hypothesis. I agree with the authors that journals should have tried to accept those studies with negative results as since "the unexpected negative results ...but for generating ideas and improving our understanding of animal color patterns more generally'. This study is the case which generates 'ideas for moving the work forward', I think. Although the study provides no evidence that male color pattern+sex-specific behavior confers an advantage to males that are proposed to imperfectly mimicking wasps, the hypotheses/predictations are very clearly spelt

out, the experiments are accordingly well designed and nicely performed, subsequent data analyses are also well done and presented. Thus, I think it is scientifically sound. Authors then provide explanations for their results, offer alternative functional hypotheses for striped color pattern of males such as disruptive and motion dazzle functions as well as even no function at all. Authors even provide two interesting insights into the biology of the species tested. I think this paper sets a stand for how to present a study with negative results for many journals, particularly those insights for moving the research forward. The manuscript is well written and presented. My only concern is that the discussion may be too long, which could be shortened one third, though many explanations could be provided for the negative results.

RESPONSE: We appreciate the positive feedback. We considered ways to shorten the Discussion, but ultimately agree with the associate editor (see comments above) that the Discussion is not overly long and so we opted not to remove additional content.

Specific Comments:

p2, line 26, Abstract: define ‘dangerous prey’ and also add ‘toxic or before ‘dangerous prey’.

RESPONSE: We use are using the term ‘dangerous prey’ broadly here, to include anything that might harm a predator in any way. This includes prey that is toxic, venomous, or otherwise could harm a predator (e.g., by biting or otherwise injuring it). We have clarified this definition in the first sentence of the Introduction, but we chose not to elaborate here to avoid exceeding the 200-word limit for the Abstract. (Note also that given this definition, we didn’t add ‘toxic or’ before ‘dangerous prey’ as the reviewer suggested; we felt that this addition did not add meaning and would result in the need to cut words elsewhere in the Abstract to remain under the 200-word limit.)

p2, line 29, Abstract: specify what are the possible hymenopteran insects.

RESPONSE: We have added “(e.g., wasps and bees)” to the Abstract to define hymenopterans for a general audience that might not be familiar with insect orders. As we clarify in the third paragraph of the Introduction, the hymenopterans that are the proposed models in this system include numerous and varied species of wasps and bees that are common in the same habitat as *H. pyrrithrix*. (The 200-word limit meant that we couldn’t also add those additional details to the Abstract).

p2, line 34, Abstract: spell out the species name of the predator.

RESPONSE: We added the species name. Note that we also adjusted wording in the rest of the Abstract to keep the word count below 200.

p3, line 49: again define ‘dangerous prey’.

RESPONSE: We added text to more clearly define what we mean by ‘dangerous prey’ (see paragraph 1 of Introduction)

p3, line 50: add a full stop ‘.’ Before ‘Selection to avoid ...’.

RESPONSE: Thank you for catching this typo – now corrected in paragraph 1 of Introduction.

P6, lines 117-118: provide the data (mean/se) on body size of both *H. pyrrithrix* (males and females) and *P. californicus* that were used in the tests.

RESPONSE: We have added these data that the reviewer suggests to the text of the manuscript (see paragraph 1 of Methods). We have also added these data to Dryad with the rest of our data.

P9, lines 184-195: is there any record of activity level for both males (2) and females (2) in each trial, which may be important for attention of predators.

RESPONSE: This is an interesting question. We did not collect data for activity rate for all of the spiders in this particular study. But we do know from field data that males spend more time moving than females (Taylor et al. 2019). The reviewer suggests that movement rates might be important for predator attention. In this case, we might expect that mobile males would be attacked more than less-mobile females but this was not the case. However, the reviewer’s comment prompted us to consider the idea that moving spiders may be less vigilant to predators (as opposed to resting spiders) and this may contribute to the higher capture rate for males. We have added this idea to paragraph 6 of Discussion.

Taylor, L. A., Cook, C., & McGraw, K. J. (2019). Variation in activity rates may explain sex-specific dorsal color patterns in *Habronattus* jumping spiders. *PloS One*, *14*(10), e0223015.

P12-15, lines 260-311: this part about possible predators for testing is too long, which can be shortened a lot.

RESPONSE: We are hesitant to remove content from this section because we feel it is necessary to generate ideas for moving this work forward. The associate editor suggested that we avoid removing content in the Discussion (see above) and Reviewer 2 suggested that the lengthy discussion of alternative predators helped alleviate concerns that we only tested a single predator. Given this, we have decided against shortening this section.

P15, lines 332-344: That females are less likely captured than males could not be due to how they are able to better avoid capture, but also may be because they are not that active.

RESPONSE: We have added a discussion of this point to paragraph 6 of the Discussion.

In addition, if the size of males and females is a possible factor, you could include body size of all 4 spiders as a predictor in the models.

RESPONSE: This is an interesting idea, but size is so tightly correlated with sex in this species (females are larger than males) so it would be impossible to tease these two factors (sex vs. size) apart. In paragraph 6 of the discussion, we discuss the idea that the difference in capture rate between males and females could be a result of the different sizes of the sexes (or various other factors that differ between males and females). We have added body size data for all of the spiders in our study to our Dryad dataset (in response to a previous comment by this reviewer above), so a curious reader is free to explore these ideas statistically.

Reviewer: 2

This is a simple, yet elegant study that addresses the question of whether dorsal patterns in a jumping spider have evolved in males as Batesian mimicry, due to males being more exposed to predation. The manuscript is well written, the objectives clearly presented, and the methods are explicit and easy to replicate. Although I personally prefer not to transform count data and instead use negative binomial regression models, the data was correctly transformed for the analyses, and so I have no comments regarding analyses. Against the predictions, the male coloration did not confer any advantage when exposed to a predator, and instead, individuals with the male markings (males and painted females) received more attacks compared to individuals without male markings (females and painted males). Regardless of coloration, males were more likely to be predated. A positive result would have provided evidence for the Batesian mimicry hypothesis, however a negative result does not provide evidence against it, as many other factors (i.e. other predators) were not included in this study. I am pleased with the way the authors acknowledge this in the discussion.

The weakness of this study is that the predation hypothesis was tested with a single predator. However, this has been thoroughly discussed, including the suggestion of specific predators for subsequent similar studies to test the Batesian mimicry hypothesis. Since jumping spiders can see in the UV spectrum, I wonder how that may have influenced the results of this study, and I would suggest the authors to expand on this in the discussion – a sentence or two would be enough.

RESPONSE: We agree that the UV spectrum should be addressed, thank you for bringing this up. We have now added text to clarify that the makeup products used have no UV peaks and closely match the spectral properties of the spider's abdominal markings. As such, there should be no unintended artifacts for predators that have UV

vision. (Note also that we direct readers to the spectral properties of these makeup products used here as they were also used in a previous study (Taylor & McGraw 2013).

Taylor, L. A., & McGraw, K. J. (2013). Male ornamental coloration improves courtship success in a jumping spider, but only in the sun. *Behavioral Ecology*, 24(4), 955-967.

In addition, the number of trials is missing in the manuscript. Since the authors collected 28 *P. californicus* spiders (predators), but one experiment was eliminated, one can easily assume that the total of the trials is 27, but in the data files it's clear that there were 13 analysed trials.

RESPONSE: Thank you for catching this. We have added the total number of trials to the 2nd paragraph of the Methods. The total was 14 trials but one trial was excluded from some of the analyses, which is already addressed later (see 3rd paragraph of statistical analysis for details).

To avoid confusion, we have corrected the total numbers of *H. pyrithrix* collected in the field to match the number of spiders actually used in our experiments (see revision to the 1st paragraph of Methods). The previous draft reported the total number of spiders collected as part of a larger study, rather than the number used in the current experiments (which was confusing).

However, it is important to note that the number of predators collected (n=28) is still higher than the total number of trials and there is a reason for this; we collected more predators than the number of trials because we wanted to ensure that we had enough predators (in paragraph 6 of the Methods, we explain that if a predator didn't attack within the first 30 minutes, the predator was removed and a new one introduced).

Other than that, I have no further comments. The authors did an excellent job in carefully editing the manuscript and I found no mistakes.

RESPONSE: Thank you, we appreciate the kind comments!